# Airspace Diameter Map—A Quantitative Measurement of All Pulmonary Airspaces to Characterize Structural Lung Diseases

**DOI:** 10.3390/cells12192375

**Published:** 2023-09-28

**Authors:** Sanja Blaskovic, Pinelopi Anagnostopoulou, Elena Borisova, Dominik Schittny, Yves Donati, David Haberthür, Zhe Zhou-Suckow, Marcus A. Mall, Christian M. Schlepütz, Marco Stampanoni, Constance Barazzone-Argiroffo, Johannes C. Schittny

**Affiliations:** 1Institute of Anatomy, University of Bern, 3012 Bern, Switzerland; sanja.blaskovic@unibe.ch (S.B.); lenabori@gmail.com (E.B.); dominik@schittny.com (D.S.); david.haberthuer@unibe.ch (D.H.); 2Medical School, University of Cyprus, Nicosia 2029, Cyprus; anagnostopoulou.pinelopi@ucy.ac.cy; 3Department of Pediatrics, Gynecology and Obstetrics, Faculty of Medicine, University of Geneva, 4 rue Gabrielle-Perret-Gentil, 1211 Genève, Switzerland; yves.donati@unige.ch (Y.D.); constance.barazzone@hcuge.ch (C.B.-A.); 4Department of Pathology and Immunology, Faculty of Medicine, University of Geneva, 1211 Geneva, Switzerland; 5Department of Translational Pulmonology, University Hospital Heidelberg, Translational Lung Research Center (TLRC), A Member of German Center for Lung Research (DZL), 69120 Heidelberg, Germany; zhe.zhou-suckow@med.uni-heidelberg.de; 6Department of Pediatric Respiratory Medicine, Immunology and Critical Care Medicine, Charité-Universitätsmedizin Berlin, 10115 Berlin, Germany; marcus.mall@charite.de; 7Berlin Institute of Health (BIH), Charité-Universitätsmedizin Berlin, 10115 Berlin, Germany; 8German Center for Lung Research (DZL), Associated Partner Site, 10115 Berlin, Germany; 9Swiss Light Source, Paul Scherrer Institute, 5232 Villigen, Switzerland; christian.schlepuetz@psi.ch (C.M.S.); stampanoni@biomed.ee.ethz.ch (M.S.); 10Institute for Biomedical Engineering, University and ETH Zürich, 8093 Zurich, Switzerland

**Keywords:** lung disease, cystic fibrosis, pulmonary emphysema, image analysis, artificial intelligence, X-ray tomographic microscopy, micro-computed X-ray tomography/micro-CT, stereology

## Abstract

(1) Background: Stereological estimations significantly contributed to our understanding of lung anatomy and physiology. Taking stereology fully 3-dimensional facilitates the estimation of novel parameters. (2) Methods: We developed a protocol for the analysis of all airspaces of an entire lung. It includes (i) high-resolution synchrotron radiation-based X-ray tomographic microscopy, (ii) image segmentation using the free machine-learning tool Ilastik and ImageJ, and (iii) calculation of the airspace diameter distribution using a diameter map function. To evaluate the new pipeline, lungs from adult mice with cystic fibrosis (CF)-like lung disease (βENaC-transgenic mice) or mice with elastase-induced emphysema were compared to healthy controls. (3) Results: We were able to show the distribution of airspace diameters throughout the entire lung, as well as separately for the conducting airways and the gas exchange area. In the pathobiological context, we observed an irregular widening of parenchymal airspaces in mice with CF-like lung disease and elastase-induced emphysema. Comparable results were obtained when analyzing lungs imaged with μCT, sugges-ting that our pipeline is applicable to different kinds of imaging modalities. (4) Conclusions: We conclude that the airspace diameter map is well suited for a detailed analysis of unevenly distri-buted structural alterations in chronic muco-obstructive lung diseases such as cystic fibrosis and COPD.

## 1. Introduction

The lung function is readily assessed with pulmonary function tests, which include spirometry, lung volume measurements, and diffusion capacity for carbon monoxide, allowing for discrimination between the healthy and the lungs affected by obstructive and restrictive damage [1]. These strategies, while very useful for the initial diagnosis, do not permit localization of the damage within the lung. Hence, they are often complemented by X-ray computed tomography (CT) or magnetic resonance imaging (MRI), the two main lung imaging techniques [2]. Both techniques have been used in clinics for more than 50 years, each with specific sets of pros and cons. While MRI provides a good soft tissue contrast without the risk of ionizing radiation, the image resolution is much lower when compared to CT, especially for the lung parenchyma [3]. Recently, additional techniques have emerged and proved important in functional and structural studies of the lung. The dark-field X-ray imaging has been suggested as a complement to CT and applied to animals and human patients with pulmonary pathologies [4,5,6,7]. This technique requires 100 times lower doses than the regular thorax CT and distinguishes well different stages of lung emphysema and pulmonary fibrosis. Furthermore, X-ray velocimetry has established itself as a functional imaging method, producing valuable data about the impact of respiratory conditions on lung function and airflow patterns [8,9,10]. Finally, several disease-specific approaches, such as PRAGMA-CF for cystic fibrosis patients or CT scoring systems of emphysema, have proven immensely valuable. These methods enable inferring lung function from CT-derived structural information [11,12,13].

To look at the lung microstructure, micro-X-ray computed tomography (µCT), with improved pixel size down to 100 nanometers, is commonly used for both animal and human studies [14,15]. A version of µCT with the highest resolution and image quality available is synchrotron radiation X-ray tomographic microscopy (SRXTM). This technique has the advantage of being operated with a monochromatic, high flux, partially coherent, and nearly parallel beam instead of the cone beam of a classical µCT, producing images with a higher signal-to-noise ratio [16]. Furthermore, depending on the application, the produced volume of the data can be vast, in which case it requires high computational power and skills as well as large storage capacity. Because of these drawbacks, the analysis of lung microstructure has been limited and often only applied to one or a few regions of the lung [17,18,19].

The extensive advance in the field of lung imaging techniques was followed by the development in the field of image analysis. Stereology has been the principal approach in lung image analysis and is crucial in addressing the structure of the homogeneous healthy lung. In fact, due to this approach, we have a broad knowledge of both the bronchial tree and gas exchange area of the lung of many different species [20]. Stereology is typically performed on systematically randomly selected lung slices and the results are then extrapolated to the entire lung. Choosing every lung section and not a sample allows an estimation of novel parameters, which require the entire 3-dimensional (3D) space.

Because healthy lung parenchyma is isotropic and homogeneous, a small sample is sufficient to give accurate results if it is properly sampled. Since, in many lung diseases, the distribution of the damage is often heterogeneous, a stereological approach becomes much more time-consuming because the fraction of the lung that has to be analyzed increases dramatically up to the entire lung. For muco-obstructive lung diseases such as cystic fibrosis (CF) and COPD, as well as interstitial lung diseases such as pulmonary fibrosis, the recommended way to properly address the extent of the damage is by analy-zing the entire lung [21,22,23,24]. While many automated and semi-automated approaches for an analysis of the entire lung have been suggested in the recent decade [25,26], no complete analysis of lung airspace distribution of the total murine lung is available to date.

In line with this, we propose a novel pipeline for the analysis of images obtained with SRXTM and µCT, which allows for an in-depth study of airspace distributions in the entire mouse and rat lung [27]. Furthermore, the pipeline can be used to extract and separately analyze any of the lung compartments. To validate our pipeline, we analyzed SRXTM images of two different lung disease models: (i) βENaC-transgenic mice, which develop a cystic fibrosis-like lung disease (CF) [28], and (ii) elastase-induced emphysema [29]. Cystic fibrosis (CF) is an inherited disease caused by a mutation in the cystic fibrosis transmembrane conductance regulator ion channel gene. This mutation disrupts the transport of sodium and chloride ions and leads to the retention of water on the surface of epithelial cells. This, in turn, results in the production of abnormally thick mucus that obstructs the conducting airways and leads to lung structural changes such as the loss of alveolar septa and an enlargement of the pulmonary airspaces resulting in a decline of lung functions [30]. Elastase-induced emphysema of mouse lungs represents a very common model for pulmonary emphysema. Pulmonary emphysema is characterized by a loss of alveolar septa (inter-airspace walls), resulting in enlarged air spaces [31].

In both cases, we compared diseased to healthy mouse lungs. We observed a clear shift towards larger size airspaces in both lung diseases as well as a completely different pattern in enlarged airspace distribution between the CF and elastase-induced emphysema. Furthermore, comparable results for the same lungs were obtained even for images with a lower signal-to-noise ratio acquired by classical µCT, suggesting that our pipeline is applicable to commonly used and more readily accessible imaging techniques. 

## 2. Materials and Methods

### 2.1. Mice

Animals were kept in specific pathogen-free facilities and exposed to 12 h day/12 h night cycles with food and drinking water ad libitum. The breading, treatment protocol (where applicable), and sacrificing of the animals were approved by the IACUC of the University of Heidelberg, Germany, for the βENaC-Tg mice and the Institutional Ethics Committee on Animal Care of the University of Geneva, Switzerland, and in accordance with the Veterinary Office of the Canton of Geneva for the C57BL/6J mice. In addition, the ARRIVE guidelines 2.0 were followed [32].

The C57BL/6J mice used for elastase/saline instillation were randomly allocated (by a coin toss), at 11 weeks of age, to either vehicle or elastase receiving group, with the prerequisite that each group contains comparable numbers of female and male mice. Elastase was administered intranasally, as described previously [29]. Briefly, for pain prevention, 0.1 mg/kg body weight buprenorphine (Temgesic, Indivior Schweiz AG, Baar, Canton of Zug, Switzerland) was admi-nistered subcutaneously 1 h before and 3 h after vehicle/elastase instillation. Vehicle or elastase (concentration of 0.2 U/g body weight; High purity porcine pancreatic elastase, #EC134, Elastin Products, Owensville, MO, USA) was instilled intranasal under 5% isoflurane anesthesia. The βENaC-Tg and the corresponding wild-type control (ctrl) mice were sacrificed at post-natal day 36 (pnd36) by an intra-peritoneal (IP) overdose of ketamine (120 mg/kg body weight) and xylazine (16 mg/kg body weight), while the elastase/saline instilled mice were sacrificed on post-elastase day (ped) 21 with an IP overdose of pentobarbital (150 mg/kg body weight). In both mouse models, the thorax was opened, and the collapsed lungs were filled with freshly prepared 4% paraformaldehyde in phosphate-buffered saline (PBS) via the trachea at a pressure of 20 cm water column [33] Afterwards, the tracheas were ligated, the lungs were removed, fixed in fresh fixative (4% paraformaldehyde) for 24 h, washed into 70% ethanol and shipped from the University of Heidelberg (βENaC-Tg mice) or the University of Geneva (C57BL/6J mice) to the University of Bern. 

Upon receiving the lungs, the tissue was post-fixed in a freshly prepared fixative. The volume of the lungs was determined by water displacement [34]. In order to achieve reliable results, the volumes of each lung lobe were measured separately instead of the entire lung. In addition, a hooked dissection needle was used instead of forceps. Afterward, the fixative was removed with PBS, and the lungs were washed stepwise into 100% ethanol and critical point dried using a Leica EM CPD 300 device (Leica Microsystems GmbH, Wetzlar, Germany) [33]. Samples were placed in a 500 μm Eppendorf tube and secured with a piece of cleansing tissue. The tubes were glued upside down on a standard scanning electron microscopy sample holder (PLANO GmbH, Wetzlar, Germany).

### 2.2. Imaging and Reconstruction of the Lung Samples

#### 2.2.1. Imaging by High-Resolution Synchrotron Radiation-Based X-ray Tomographic Microscopy (SRXTM) at the TOMCAT Beamline

The TOMCAT beamline (Tomographic Microscopy and Coherent radiology experiments, X02DA) at the Swiss Light Source (Paul Scherrer Institute, Villigen, Switzerland) was used for image acquisition with a monochromatic X-ray beam at an energy of 12 keV. To record the images, we used a high numerical aperture white-beam compatible 4× ma-croscope (Optique Peter, Lentilly, France) equipped with a pco.edge 5.5 sCMOS camera (PCO GmbH, Kehlheim, Germany) and a 17 μm thick LSO:Tb (E17LSb) scintillator (Crytur, Turnov, Czech Republic), yielding a voxel size of (1.625 µm^3^) and a field of view of 4.2 × 3.5 mm^2^. Three samples per condition were imaged using a protocol of stacked 360° scans. In brief, to increase the field of view in parallel to the rotational axis, 4–7 scans were taken (stacked) on top of each other. To extend the field of view by a factor of ca. 1.9 perpendicularly to the rotational axis, the rotational axis was moved from the middle of the field of view (180° scan) close to the right or left border, and the sample was turned 360° instead of 180°. Before reconstruction, the resulting projections were stitched to their corresponding projections (0° images to the reversed 180° image, etc.) in order to achieve a “virtual” projection of nearly twice the field of view of the camera. The total scan time was 21–42 min/lung for the 36-day-old mice (βENaC-Tg and the controls) and 41–48 min/lung for the 100-day-old mice (elastase-instilled and the ctrl’s). The number of projections was set to 3001/per 360° scan at an exposure time of 100 ms [27].

The reconstructions of individual 3D datasets were performed as described previously in detail in [27]. In brief, we first applied dark-field correction using the mean of 30 calibration dark frames, followed by a dynamic flat-field correction [35]. To increase the air-tissue contrast, the projections were submitted to a phase retrieval process using a method developed by Paganin et al. [36] with δ and β parameters set to 2 × 10^−7^ and 2.8 × 10^−10^ and the propagation distance of 50 mm. Subsequently, the projections were reconstructed using the gridrec algorithm with recorded angular positions and center offsets [37]. The rotation center of individual samples was determined manually by screening multiple rotation axis positions.

#### 2.2.2. μCT Scans

A large-capacity 3D X-ray microscope (Bruker SkyScan 1272 (Control software v1.1.19, Bruker microCT, Kontich, Belgium)), equipped with a Hamamatsu L11871_20 X-ray source and a XIMEA xiRAY16 camera, was used to image the left lobes of one ctrl and two elastase-instilled lungs. The X-ray source was set to a tube voltage of 50.0 kV and a tube current of 200.0 μA; the X-ray spectrum was not filtered. We recorded a set of three stacked scans overlapping the total height of each sample. Two samples (one control (ctrl) and one emphysema (El-1), previously used in SRXTM experiments) were recorded with a camera size of 4904 × 3280 pixels, a frame averaging 3, and an exposure time of 1331 ms per projection. The large lateral extent of the third emphysema sample (El-2, additional sample, not used previously in SRXTM experiments) necessitated recording it with so-called stitched projections, with a (virtual) camera size of 9328 × 3277 pixels. This sample was recorded with an exposure time of 800 ms per projection. All samples were scanned with approx. One thousand nine hundred projections at every 0.1° step over a 180° sample rotation. The total scan duration was about 3 h and 50 min per stack for the ctrl and El-1 sample and 5 h and 39 m for the El-2 sample. This resulted in a total scanning time of approximately 40 h. The projection images were then subsequently reconstructed into a 3D stack of images with NRecon (v1.7.4.2, Bruker microCT, Kontich, Belgium) using a ring artifact correction of 14, yielding 8-bit files with an isometric voxel size of 1.5 μm.

### 2.3. Image Stitching and Analysis

#### 2.3.1. Stitching of Individual SRXTM Scans

The individual data volumes (from 4 to 7 blocks) were assembled into one z-stack with the non-rigid stitching framework NRStitcher [27]. The final stitched dataset size for the βENaC-Tg lungs and the day 36 controls were in the range between 231 and 805 GB with width x length x height dimensions of 4100–5700 × 4300–6396 × 6070–10740 voxels, which corresponds to a physical volume of 6.7–9.3 × 7.0–10.4 × 9.9–17.5 mm^3^. The final stitched dataset size for the elastase/saline-instilled lungs was a 16-bit image in the range between 401 and 609 GB with dimensions of 4100–5000 × 4100–4600 × 11837–14230 voxels, corresponding to a physical volume of 6.7–8.1 × 6.7–7.5 × 19.2–23.1 mm^3^. The illustration of the output of image reconstruction and stitching is shown in Figure 1B,C.

#### 2.3.2. Statistical Analysis of the Results

In order to compare the histograms of βENaC-Tg/elastase-instilled and control samples the following parameters of the airspace diameter distribution fit curve were compared: (i) peak position and height (Figure 2A), (ii) at half peak height the peak width and the position of the left and right shoulders (Figure 2B), and (iii) the area under the curve left and right of the crossing point of the two plots (e.g., βENaC-Tg and control, as well as elastase-instilled and control) (Figure 2C,D). Because no data point was located exactly at the half-peak height, the two closed data points and linear interpolation were used. The area under the curve is a direct measure of the airspace volumes and can be converted to the airspace volume by the following formula: (area under the curve multiplied by voxel size (1.625^3^))/shrinkage factor.

The student’s *t*-test, two-tailed and homoscedastic, was performed to determine the significance of the difference between the ctrl and diseased lungs of the above-mentioned parameters.

#### 2.3.3. Plotting and Visualization of the Distribution of Enlarged Airspaces

The results were plotted with the R project (v4.1.0) and ggplot2 package, and the data were fitted with the generalized additive model (GAM) with integrated smoothness estimation with the following formula y~s (x, k = 50) (mgcv package), with k representing the degrees of freedom [38,39,40,41]. For a visual illustration, we used Imaris software (v9.3.1, Bitplane, Zürich, Switzerland). To account for the increased lung volume in both disease models, we presented the raw counts together with normalized probability curves (count per diameter divided by the sum of all counts).

## 3. Results

### 3.1. Segmentations

#### 3.1.1. Segmentation of Pulmonary Tissue

Samples were imaged in 3D by SRXTM (see above, Section 2.2.1) and/or with high-resolution μCT (see above, Section 2.2.2). Reconstructed images were further processed with an open-source machine-learning tool, Ilastik (v1.3.3) [42]. This approach was chosen due to the high level of background gradient detected in the samples. We used a pixel classification workflow that extracts different pixel features (pixel intensity, pixel boundaries, and variations in pixel values within a neighborhood) from images for the purpose of classification. The scale of the applied feature can be tuned by a sigma factor, where smaller sigma values capture finer details, while larger values capture broader structures. Furthermore, the workflow can operate in two-dimensional (2D) or three-dimensional (3D) space. In our study the following pixel classification workflow was used: (i) Gaussian smoothing was used to detect pixel intensity with the sigma values of 0.3 and 1 in 3D and 3.5 and 10 in 2D; (ii) pixel boundaries were detected with three edge filters (Laplacian of Gaussian, Gaussian Gradient Magnitude and Difference of Gaussians) with the sigma values of 0.7 and 1.6 in 3D and 5 in 2D, and (iii) pixel variations within a neighborhood were detected with two texture filters (Structure Tensor Eigenvalues and Hessian of Gaussian Eigenvalues) with the sigma values of 1 in 3D and 3.5 and 10 in 2D. The Ilastik projects were trained on 12 consecutive slices at three different positions in the z-stack to best re-present all the different background gradients within the z-stack. Following the features selection process, a manual input was given in order to label the lung tissue and separate it from the rest of the image (lung airspaces and the background surrounding the lung). Due to the large amount of data, the object classification by Ilastik was performed in “headless” mode, which processes the images by dividing them into independent blocks. The amount of trained Ilastik projects depends on the similarities of the background and tissue intensities between samples. If two samples have similar backgrounds and tissue intensities, then the same project can be applied to segment both samples. For the images obtained by SRXTM, each of the samples in the βENaC-Tg group was segmented with a different Ilastik project, and for the samples in the emphysema group, we used only 2 projects for segmentation of the controls and only 1 project for segmentation of all the elastase-instilled samples. For the images obtained with the μCT, each of the three samples was segmented with a different project. The illustration of the Ilastik segmentation output is shown in (Figure 1D). The segmentation of every lung sample was judged by a trained stereologist by asking if the voxels were properly classified as tissue or airspace. 

#### 3.1.2. Segmentation of Pulmonary Airspaces

To obtain a segmentation of the pulmonary airspaces, we first created the mask of all lung lobes (tissues and airspaces). First, a tissue “dilation” was applied to the segmentation of the pulmonary tissue. A radius of five pixels was used to close small holes in the alveolar septa. Second, the “fill holes” function of ImageJ was applied to the “dilated” dataset [43]. Third, the obtained mask of the lung lobes was eroded by the same radius as applied in the first step (dilation). 

The final segmentation of pulmonary airspaces was created by multiplying the inverted result of segmentation of the pulmonary tissue with the mask of lung lobes. The output of the segmentation process is a binary image with the same dimensions as the input image obtained after the stitching step. For the analysis of the µCT images, the final segmentation file was further processed with the “analyze particles” function of ImageJ by applying a filter size of 50 pixel^2^, meaning that all the particles whose area is smaller than 50 pixel^2^ (e.g., in the case of a perfect circle, all the particles with the radius smaller than 4 pixels) will be removed. This “cleaning” step was required because of the higher background level in the images obtained with the µCT. An illustration of the final segmentation is shown in Figure 1E.

### 3.2. Calculation of Airspace Diameter Distribution Using the Thickness Map Algorithm

The algorithm for the calculations of airspace diameter distribution was originally developed for bone [44] and material science [45] and applied to the lung as described previously [46]. Briefly, spheres of maximal diameter were fitted into the 3-dimensional lung airspaces using the “thickness map” function of the image analysis program Pi2 (https://github.com/arttumiettinen/pi2 (accessed on 30 March 2022)) (Figure 3).

The airway diameters were extracted and mapped as grey values of every voxel. An illustration of the airway diameter map is shown in Figure 1F. Using the ImageJ (v1.54f) function “Histogram” with the 150 bins of a bin size of 2 and x value range from 3 to 303, a histogram distribution was calculated showing the number of voxels of the certain grey value or the airway diameter, respectively. 

To normalize the uneven volume between individual lungs, we converted voxel counts into voxel probability by dividing the count for the determined diameter by the total voxel count of the lung. Finally, to account for the tissue shrinkage due to critical point drying, we compared lung volume obtained by water displacement with the volume obtained from the mask of all lung lobes (described in Section 3.1.2). We observed an aver-age shrinkage of the sample volume of 70.6% (+/−1.2%). This resulted in a mean volume of 29.4% (shrinkage factor) of the original volume. These values were not far from the values published for the rat lung shrinkage, where the shrinking factor was estimated to be 36.8% [47]. We corrected all our measurements for shrinkage: (i) for the length mea-surements (e.g., airspace diameter) by dividing the value with ^3^√shrinkage factor, (ii) for the alveolar surface measurements by dividing the value with ^2^√shrinkage factor and for the (iii) volume measurements, by dividing the value with the shrinkage factor. 

### 3.3. Extraction of the Conducting Airways

The bronchial tree of the lungs may be divided into two parts. The conducting and the gas-exchanging airways. The conducting airways consist of the bronchi (larger conducting airways containing cartilage in their walls) and the bronchioles (smaller conducting airways, no cartilage present). Every of the most distal bronchioles (terminal bronchioles) connects to a so-called acinus, which represents a small tree of gas-exchanging airways. The total volume of all acini is equal to the lung parenchyma or gas exchange area, respectively [48].

In 3D visualizations of the airway diameter map of the bronchial tree, the transition from the bronchioles to the airways of the acini (alveolar ducts) is easily detected by the appearance of the alveoli in the walls of the alveolar ducts (Figure 4A, red circles). Based on this criterion, we observed that the diameter of the terminal bronchioles is quite constant in a particular animal. This diameter increases during lung development but is still quite constant on a given day of postnatal lung development. However, we observed small differences between different animals of the same age. 

We used this morphologic criterion to find the threshold of the airway diameter, which separates conducting and gas-exchanging airways (Figure 4). By this definition, all voxels of the conducting airways show larger gray values (encoded airway diameter) than the threshold. We used this definition to extract the conducting airways out of the airway diameter map of an entire lung or lung lobes, respectively. Hence, this definition is not exactly the same as the classical definition that the conducting airways end with the terminal bronchioles (for a review, see [48]) because of a small but significant variation in the diameter of the terminal bronchioles. Therefore, based on this morphological definition, some of the terminal bronchioles may be missed, and some of the most proximal alveolar ducts may be included in our extraction of the conducting airways (Figure 4). 

To extract the conducting airways (CA), we first applied the above-mentioned thres-hold on the airway diameter map and discarded all the lower grey values (smaller airway diameters). To discard all the enlarged airspaces in the βENaC-Tg lungs that are not part of the conducting airways but share the same diameter value, we performed the step of connected component analysis that consists of two parts: (a) selecting a pixel with the coordinates (x, y, z) within the conducting airways and (b) growing this pixel into all neighboring pixels that are connected to it and have the same intensity value in 3D. Finally, the created mask of the conducting airways was multiplied with the original thickness map to extract the airway diameter map. To obtain the count distribution for the gas exchange area (lung parenchyma), we subtracted the counts of the conducting airways from the total lung counts. The above processing was performed with the help of the Pi2 package. The representative 3D visualization of the entire lung and the two different compartments is shown in Figure 5A, while the plots for count/probability distribution of the conducting airways and the gas exchange area are shown in Figure 5B–E. Finally, the statistics are summarized in Table 1.

As expected, the diameters of the conducting airways were much larger than the gas exchange area, with the most frequent airspace diameter being 6× larger (122 vs. 22 μm, Table 1). The total airspace volume, on the other hand, is 12× bigger for the gas exchange area (3.1 × 10^10^ vs. 3.5 × 10^11^, Table 1), which means that the conducting airways airspaces represent 8% of the total lung airspace volume. 

### 3.4. Airspace Diameter Map of the βENaC-Tg Lungs Scanned by SRXTM

The β-ENaC-transgenic mice represent an animal model for CF-like disease. We utilized this model and a mouse model for chronic obstructive pulmonary disease [49,50] (see below, Section 3.5) to demonstrate the analytical potential of the airspace diameter map.

The image analysis was performed on the whole lung scans obtained by SRXTM-imaging of controls (ctrl) and βENaC-Tg lungs from pnd36. The visual illustration of a representative lung slice and the results of image analysis are shown in Figure 6, while the statistics are summarized in Table 2. The accuracy of each individual fit can be verified in our Appendix A.

The peak position of the fits, shown in Figure 6, represents the most frequent airspace found in the lung and is 60% larger in βENaC-Tg lungs when compared to the ctrl (Table 2). The total lung airspace volume, obtained with image analysis, is also doubled in βENaC-Tg lungs (0.66 vs. 0.32 mL, Table 2, Volume of airspaces). 

The peak position and peak width at half height are well-established characteristics for the comparison of the plots of two histograms (Table 2). To compare the shoulders of the peak, which are lower than the half peak heights, we compared the area under the curve between ctrl and the diseased lungs in addition. Because the two curves cross, we had to compare the area before and after the intersection. While the areas before the intersection are comparable for both the ctrl and βENaC-Tg lungs, the area under the curve measured after the intersection is 3.6 times larger in the βENaC-Tg group (Table 2).

To account for the increase in the total volume in βENaC-Tg lungs, we represented our results not only as raw counts (Figure 6B,C), but also as probability curves (Figure 6D,E). The statistical analysis performed on the probability curves further emphasizes the difference between βENaC-Tg and ctrl lungs (Table 2). After this normalization, the difference in the area under the curve becomes significant between the two groups, both before and after the curve’s intersection, indicating that all size airspaces are affected in the CF-like disease.

Finally, to visualize the difference observed in the fit curve shoulders between the ctrl and βENaC-Tg lungs in Figure 6B–E, we extracted the gas exchange area with diameter sizes ranging between 80 and 140 μm and merged it to a visualization of the airspace diameter map of the entire lung (Figure 7).

We (i) observed the gas exchange area with diameters ranging from 80 to 140 μm is much more abundant in the βENaC-Tg lungs than in controls, and (ii) we visualize the actual localization of this portion of the gas exchange area. By doing so, it is possible to detect disease-induced morphological differences in isolated and small regions of the lung. Furthermore, we observed an uneven distribution of airspaces possessing a diameter of 80 to 140 μm in the control lungs. This class of airspaces was most abundant in the first generations of the alveolar ducts of acini located below the pleura at the edges of the lung. 

### 3.5. Airspace Diameter Map of Emphysematous Lungs Scanned by SRXTM

To assess the airspace volume in the ctrl and elastase-instilled adult mice lungs, we performed the same analysis as for the βENaC-Tg lungs (see above). The results are shown in Figure 8A–E and Table 3, and the accuracy of each fit can be verified in our Appendix A. Here, the most frequent airspace is the same in both groups, with a diameter of 27 μm. However, the total airspace volume is 50% larger in elastase-instilled lungs (0.63 ± 0.04 vs. 0.42 ± 0.04 mL). This enlargement cannot exclusively be explained by a simple increase in total lung volume, as the elastase-instilled lungs are only 21% larger than the controls (0.86 ± 0.04 vs. 0.71 ± 0.04 mL, measured by water displacement technique). Most likely, the loss of alveolar septa has also contributed to the increase of the airspace volume.

To better assess the distribution curve, we divided the curve again into two parts: before and from the shoulder. We observed that the smaller size airspaces are not significantly different between the control and elastase-instilled lungs when looking at the total counts (surface area and volume before the shoulder). On the other hand, the volume of larger airspaces (the shoulder until the end, starting with a diameter of 46 μm) is 2.5 times larger in the elastase-instilled lungs. 

Because the elastase-instilled lungs are 21% bigger, we present our results in the form of probability distributions normalized with total lung count (Figure 8D,E). After this normalization, the area under the curve before the shoulder is now significantly lower, while the area in the second part of the curve stays significantly larger in the emphysematic lungs, confirming once more the existence of enlarged airspaces in the elastase-instilled lungs.

### 3.6. Comparison of Image Analysis Results of Emphysematous Lungs Scanned by SRXTM and CT—The Impact of Data Quality on Image Analysis Output

Our strategy has provided reproducible results throughout all the samples scanned with synchrotron-based X-ray tomography. However, due to the lower accessibility of SRXTM, the μCT is still by far a much more common imaging technique. We therefore aimed to see if our approach is applicable to images obtained by a μCT. For this purpose, we imaged three left lungs with 2 different exposure times by μCT: one control sample (ctrl) and one elastase-instilled sample (El-1) with extensive emphysema were scanned with an exposure time of 1331 ms (yielding images with lower noise) and one elastase-instilled sample (El-2), with only very mild emphysema, scanned with an exposure time of 800 ms (yielding images with higher background).

The representative images and the corresponding results of image analyses, taken by SRXTM or µCT, are shown in Figure 9. The statistics are summarized in Table 4. As expected, µCT images (Figure 9A, right panels) had an increased background with many white speckles and lower resolution of the lung parenchyma. 

To compare the results scanned by two different tomographic imaging setups, we had to take into account the different voxel side lengths (1.5 μm versus 1.625 μm). For the same volume, the total number of voxels is 1.27 times larger in the μCT-datasets as compared to SRXTM. To circumvent this difference, we compared probabilities only. No relevant differences between the datasets scanned by SRXTM or μCT were observed for the samples scanned with higher CT resolution (ctrl + El-1; 1331 μs exposure time; Figure 9B–D, Table 4). However, in the “low-resolution” μCT scan of sample El-2 (800 μs exposure time), we did not detect the enlarged airspaces to the same extent as in the high-resolution SRXTM scans. Most likely, the larger airspaces were not detected by our algorithm due to the remaining background falsely defined as tissue. 

## 4. Discussion

We are presenting a novel pipeline for segmentation of pulmonary airspaces and for an estimation of airspaces diameters and volumes of an entire murine lung from images acquired with synchrotron radiation-based X-ray tomographic microscopy (SRXTM) and conventional micro-X-ray computed tomography (µCT). The steps are performed with entirely free (with the exception of Imaris) and user-friendly software and toolkits. Our pipeline can be used (i) to address the distribution of airspaces throughout the entire lung and (ii) to extract and separately analyze different lung compartments. We call the results of this pipeline an “airspace diameter map”. 

This type of in-depth analysis of the entire lung is particularly important for the investigation of diseased lungs, as the damage is often patchy. A typical example of a patchy distribution is our model of elastase-instilled lung. In this model, the airspace destruction is a consequence of the model itself and how the elastase enzyme is administered into the lung. Namely, not all the airspaces will encounter elastase, and only the ones that do will be modified by the enzyme. These lungs show an uneven, patchy distribution of diseased and healthy areas. Their airspace diameter probability distributions (Figure 8A,D,E, as well as Figure 9A) still show the same peak position as the controls, indicating the pre-sence of regular, healthy-sized airspaces. However, we can also appreciate that the size distribution curve for these animals has a shoulder in the range of larger diameter values, which is absent in the control. This shoulder represents the enlarged, diseased airspaces. 

The mean linear intercept (MLI) is based on the mean number of intersections of lung tissue with test lines. It is best characterized as a stereological estimation of mean free distances between gas exchange surfaces in the lung parenchyma. It is commonly used to determine the extent of emphysema in the lungs. MLI represents the closest stereological estimation as compared to the airway diameter map but is still a completely different parameter because (i) the airway diameter map determines the large ball fitting into the airspace and not the distance between two airspaces and (ii) the airway diameter map looks to the distribution of individual airspaces and not to a mean. To link the airspace diameter map to a stereological parameter, we would like to compare the two entities anyhow.

There are several publications that tried to address the damage provoked by elastase instillation in C57BL/6J mice by a stereological analysis and calculation of the MLI. However, there is a large discrepancy between the results obtained for MLI in these studies, with values ranging between 40 and 115 μm [51,52,53]. It may be explained by an inhomogeneous distribution of the emphysema and other differences between the studies. To exclude the first possible reason, the estimation of an alternative parameter would be desirable. While we cannot claim that the pipeline suggested in our study would do a better job, it is reasonable to assume that the analysis of 100% of a sample in our case, as compared to less than 5% in stereology, would give more reliable results. 

Interestingly, in the βENaC-Tg mice, we encountered a very different airspace dia-meter distribution as compared to elastase-instilled lungs. While both models share the common feature of enlarged pulmonary airspaces, in the βENaC-Tg lungs, all airspaces are affected by the disease, and the peak of the airspace diameter distribution is moved towards a larger diameter. 

βENaC-Tg lungs not only show an excess of elastase but also have mucus plugs. These plugs eventually lead to increased pressure and physical forces that, together with the progressive loss of elastic recoil, loss of alveolar septa, and complex epithelial remo-deling, contribute to the enlargement of the pulmonary airspaces [54,55,56]. 

Morphological inspection of the lung slices of βENaC-Tg lungs and elastase-instilled lungs revealed that the damage is more homogeneously distributed in the βENaC-Tg lungs than in elastase-instilled ones (Figure 6, Figure 7, Figure 8 and Figure 9). Therefore, the mean alone is a representative value for the whole βENaC-Tg lung. In this case, the mean airspace diameter value obtained for the control mice was 60.4 ± 0.6 µm for day 36 and 60.2 ± 1.6 µm for day 100. These results correlate well with the published MLI values [18], even if a direct comparison is not possible (see above). For the mean airspace diameter in the βENaC-Tg lungs, we did not find any publication addressing exactly the same age group as in our paper. However, the data published for MLI for days 14 and 60 are 75 and 80 µm, which is comparable to our results for day 36 (78.75 ± 2.9 µm) [57]. 

To analyze the different lung compartments, we have used the connected component analysis and a 3D visualization tool. This approach allows us to perform quantitative ana-lysis to address either lung development or changes occurring in the disease in the particular region of the lung. In the example presented in this paper, we compared the airspace diameter distribution between the conducting airways and the gas exchange area. The threshold for the conducting airways was chosen based on morphological criteria and did not involve the counting of generations and bifurcations. In this sense, our approach is less accurate than the conducting airways model proposed by Weibel [58]. Because we extract exclusively based on airspace diameter, in some of the places, we will have regions that are already in the acini, and in others, we might not arrive at the end of the most distal generation of bronchioles. The challenge is to manually select a threshold that will represent the best compromise between the two. Nevertheless, the threshold we chose as correct in this study was the previously published lowest diameter (100 μm) of the terminal bronchioles [59] and in the range published in the 2nd study [60] for the same mice background and age 7–8 weeks. 

As expected, our analysis showed a completely different distribution between the two anatomically distinguished pulmonary regions. The airspace diameter distribution in the gas exchange area (diameter sizes~7 to 100 µm) was represented with a smooth curve and a single peak at the diameter of 22 µm. On the other hand, the airspace diameter distribution of the conducting airways shows a peak at a much larger diameter value (122 µm), followed by several smaller peaks. The reason for the presence of these “waves” lies in the structure of the conducting airways. Namely, before every bifurcation, there is a widening of the airway, which corresponds to the waves detected in the graph. As the level of widening is not constant and varies between different bifurcations, the “waves” of the curve are quite irregular (Figure 5C,E).

The approach of extraction of the conducting airways or gas exchange area can be applied to assess any other part of the lung. For example, the structure of different parts of the conducting airways. This area has been vastly studied since the fifties of the last century and in many different species [58,61,62]. Researchers have used common parameters such as the number of airways per generation the diameter and length of the branches, as well as the branching angles in order to describe its structure [20]. While some basic concepts are well established, for example, the presence of dichotomous branching in humans or the variability of the degree of asymmetry of bifurcations between different species and between central and peripheral areas of the same lung, many aspects of this complex question remain open and important to address for better understanding of gas flow and particle deposition. In line with this, many potential applications of our approach could be envisaged, one being a systematic comparison of the airspace diameter distribution between different generations of bronchi within a single animal, amongst different animals, or, finally, between healthy and diseased animals. 

Furthermore, a specific range of airspace diameters belonging to the lung parenchyma may be extracted, visualized, and analyzed, as shown in Figure 7. Figure 7 visualizes the airspace widening observed in the βENaC-Tg lungs in comparison to controls, as well as a specific appearance of widened airspaces in the first generations of the alveolar ducts located at the edges of the lung. It lies beyond the scope of this manuscript to analyze these differences. The aim is to show the potential of further studies.

As with any approach, the one proposed here also has certain limitations: (i) because it requires a relatively high image quality, the field of view is limited, and currently, imaging is only possible for small animals, and (ii) the requirement for large data storage and analysis resources. Furthermore, (iii) the experiment was performed on excised lungs rather than on live animals. While different method exists for sample preparation in order to prevent degradation and motion during the image acquisition, it is reasonable to expect some alteration in the lung microstructure due to the change in surface tension. The critical point drying used here causes sustainable shrinkage that needs to be accounted for. 

Finally, the most important aspect of our pipeline is that it can be applied to 3D datasets acquired not only with SRXTM but also with the regular µCT, which, due to its higher accessibility, remains the more frequently used x-ray-based imaging method. This was not intuitive as these two imaging techniques differ significantly in signal-to-noise ratio due to different types of beams used for sample illumination: (i) the SRXTM uses a parallel beam that does not diverge significantly as it passes through the sample and allows for more efficient use of photons. On the other hand, the µCT uses a cone beam, which results in an uneven illumination of the sample and only a fraction of the photons being captured by the detector, and (ii) the SRXTM uses a monochromatic beam (very narrow band of wavelengths) while the µCT uses a white beam (very wide band of wavelengths). Furthermore, in our analysis, the output files for the µCT were 8-bit vs. 16-bit images in SRXTM, which resulted in an easier and more accurate segmentation for the SRXTM images.

The higher noise level of the µCT dataset was visible in the form of white speckles found also within the pulmonary airspaces. Unfortunately, some of these “background speckles” remained even after the application of the “analyze particles” function. This was because the function was not able to distinguish between the large speckles and alveolar septa (see Section 3.1.2). As the diameter of the airspaces is calculated by fitting a sphere of maximal diameter into the airspace, the presence of the speckles cuts the available space and results in the fitting of smaller spheres, thus causing an underestimation of the true diameter. This problem was especially visible when analyzing the sample El-2, which possessed a higher noise level. Namely, the analysis of this sample showed a 50% decrease in the volume belonging to the curve’s shoulder, which represents the elastase-induced enlarged airspaces. Therefore, while our pipeline is applicable to the μCT images, it is important to use images with an optimal signal-to-noise ratio to obtain accurate and reproducible results.

An additional aspect to consider when comparing two different techniques is the importance of pixel size adjustment. Because we used 2 different pixel sizes (1.625 µm for SRXTM scans and 1.5 µm for the µCT scans), we observed a discrepancy in the total voxel count, which was 30% higher for the µCT data. This issue can, however, be corrected by normalizing the individual counts per grayscale value to the sum of total pulmonary airspace counts and representing the results as a histogram of probabilities. Finally, there is a significant difference in time spent on acquiring the images. In our case, the duration of the scan with the µCT was 11.5 h for only the left lung (which is about 35% of total lung volume), while the duration of the SRXTM scan was around 45 min per whole lung. On the other hand, µCT is much more accessible than SRXTM. 

## 5. Conclusions

In conclusion, we established a novel pipeline for lung image analysis of small animals acquired by X-ray-based high-resolution μCT. The pipeline consists of steps that are performed with free and user-friendly software and allow for the analysis of airspace dia-meter distribution within the whole lung and dissection of different lung compartments. Furthermore, as proof of our pipeline efficiency, we compared the whole lung airspace diameter distribution in the healthy mice with CF-like lung disease and/or elastase-induced emphysema and found a clear separation of distributions between the healthy and diseased subjects. This work is important because (i) it contains the distribution of total lung airspaces, rather than averages as obtained by classical methods, (ii) it suggests an automated way to separately analyze different lung compartments, which is critical for better understanding the mechanism of inhalation and particle deposition of harmful and therapeutic substances.

## Figures and Tables

**Figure 1 cells-12-02375-f001:**
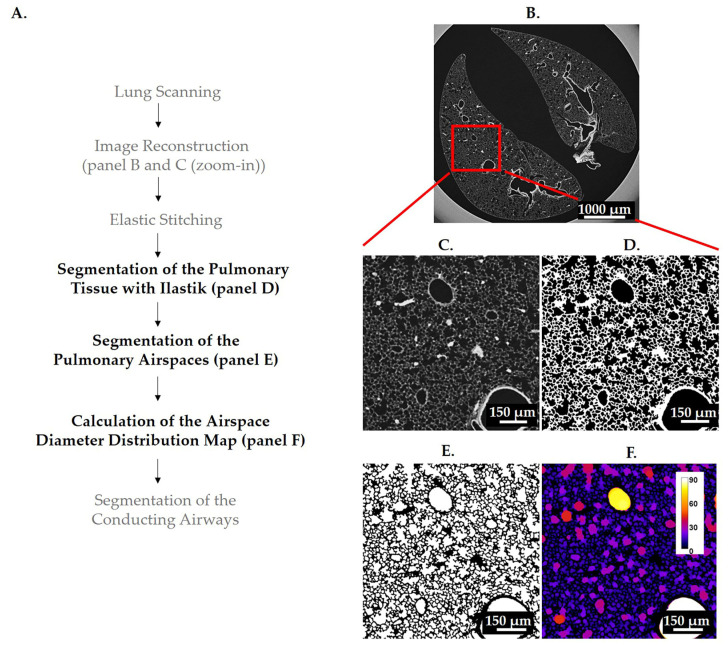
Illustration of image processing and analysis pipeline. A flowchart of all steps used in image acquisition, processing, and analysis is shown in (**A**). The steps involved in the image analysis pipeline are labeled with bold, black letters, while the other steps are labeled with regular, gray letters. A control lung from pnd36 was used to illustrate the results of image reconstruction (**B**,**C**) and the results of steps involved in the lung analysis pipeline (**D**–**F**). One lung slice is shown at low (**B**) and high (**C**) magnification (zoom in view, red square). The pulmonary tissue was first segmented with Ilastik (**D**) and subsequently used to obtain the segmentation of pulmonary airspaces (**E**). The distribution of airspace diameters was determined 3-dimensionally with the thickness map algorithm and is shown as a heatmap where the brighter colors correspond to the larger regions (**F**). The color code for the airspace diameter is shown with the calibration bar (**F**).

**Figure 2 cells-12-02375-f002:**
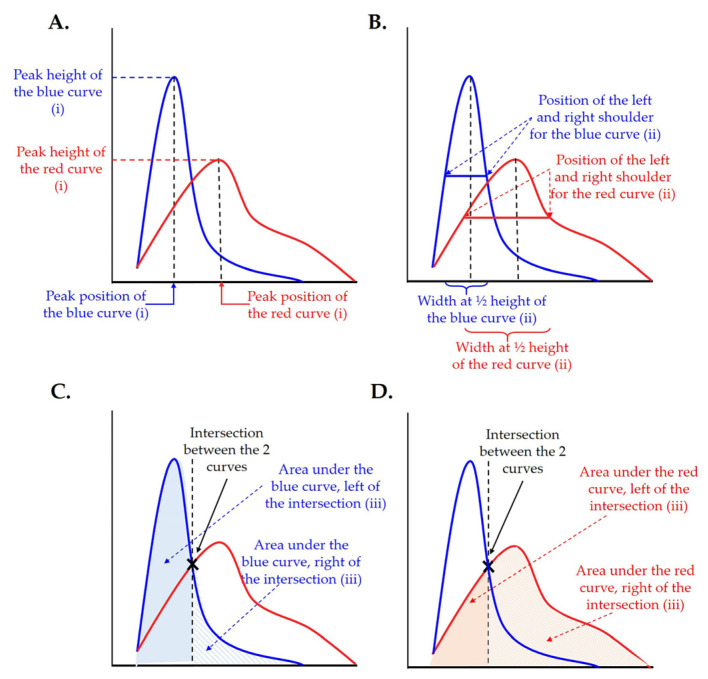
Illustration of the parameters used in the statistical analysis. Different parameters of fit curves were assessed to determine the difference between βENaC-Tg/elastase-instilled- and control samples: peak height and peak position (**A**), peak width and the position of the left and right shoulders at half peak height (**B**), and area under the curve left and right of the intersection (**C**,**D**). The latter compares the raising of the curves left of the intersection and the shoulders right of the intersections.

**Figure 3 cells-12-02375-f003:**
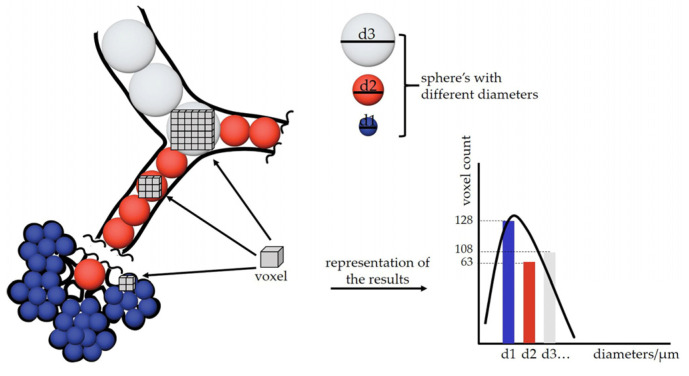
Schematic representation of an airway diameter map. At every voxel, a sphere with the maximal diameter is fitted into the airspace (different diameter spheres are represented with different colors: blue, orange, and white). Afterward, a gray value that is equal to the diameter of the sphere is assigned to the voxel. We gain a map where the particular airspace diameter is encoded in every voxel. Voxel-counting is performed for every gray value and presented as a histogram, where the “number of voxels” (voxel count) is plotted over the “diameter of the airspace where the voxels are residing in”.

**Figure 4 cells-12-02375-f004:**
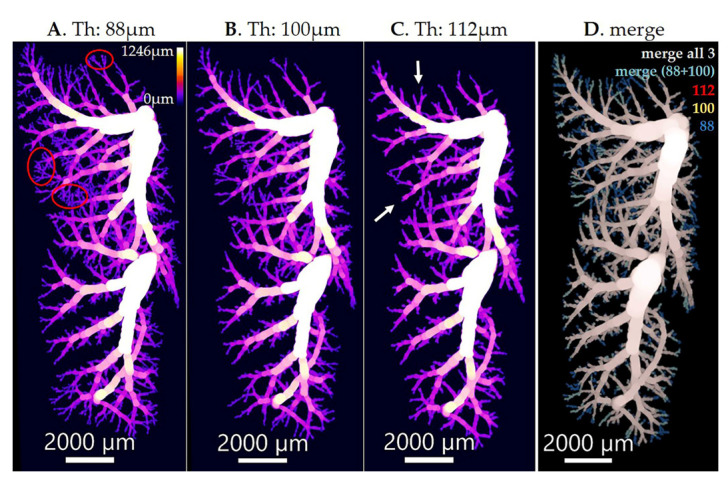
Imaris illustration of conducting airways. The conducting airways, isolated with a diameter threshold of 88 μm (**A**), 100 μm (**B**), and 112 μm (**C**), were visualized, and the overlap between three different thresholds is shown in (**D**). The threshold of 100 μm was chosen as the correct thres-hold. By applying a threshold that is too low, e.g., 88 μm, we obtain the alveoli together with the conducting airways (**A**, red circles show the alveoli). On the other hand, by applying a threshold that is too high, e.g., 112 μm, we are missing some parts of the conducting airways (**C**, white arrows show the location of missing parts of the conducting airways). The scale bar is shown in every image on the bottom, while the color bar indicating diameter sizes in μm is shown in the top right corner of the image (**A**). The color of each channel in (**D**) is shown in the upper right corner (112 in red, 100 in yellow, and 88 in blue), and the merges are indicated at the top of the image (white/gray for the merge of all three thresholds and turquoise for the merge of threshold 88 and 100). Threshold (Th).

**Figure 5 cells-12-02375-f005:**
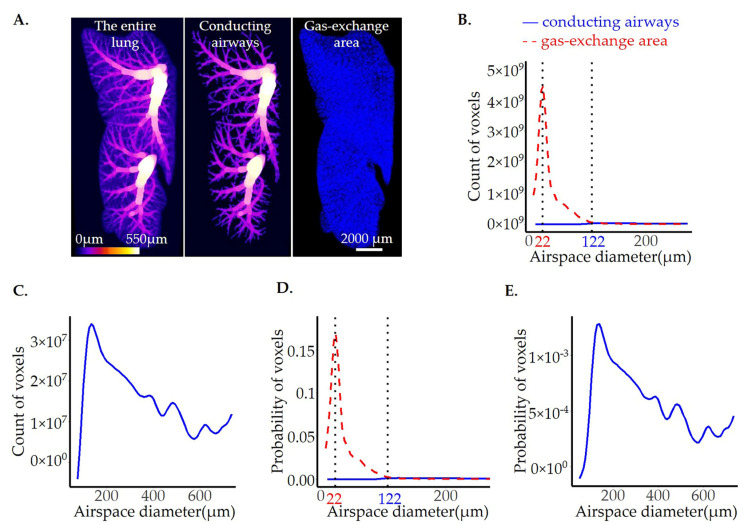
Separation of the gas exchange area from the conducting airways in healthy mouse lung from pnd36. The illustration of the entire lung (**A**, left panel) and different lung compartments, conducting airways (**A**, middle panel), and gas exchange area (**A**, right panel) were prepared in Imaris. The calibration bar showing the color code for the airspace diameter sizes is shown in the bottom left corner of the left panel, and the scale bar showing the image size is shown in the bottom right corner of the right panel). The thickness of airspaces residing in gas exchange areas and conducting airways is shown as fits of histogram distributions of voxel counts (**B**,**C**) and probabilities (**D**,**E**) over airspace diameter (µm). The results are shown with a solid blue line for conducting airways and a dashed red line for the gas exchange area. As the curves for conducting airways are not well visible in graphs (**B**,**D**) due to the y-scale range, the zoom-in into conducting airways curves are shown in (**C**,**E**). The position of the maximal curve’s peak, representing the most frequent airspace diameter, is shown with the black dotted line, and the value of airspace diameter associated with the peak is labeled in blue for the conducting airways and red for the gas exchange area.

**Figure 6 cells-12-02375-f006:**
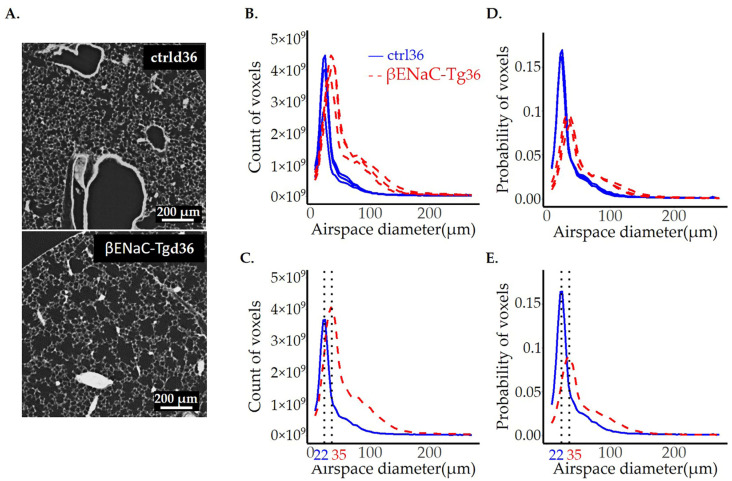
Comparison of control and βENaC-Tg mice lungs with CF-like disease from pnd36. One representative lung slice from the ctrl (**A**, upper panel) and βENaC-Tg (**A**, lower panel) mice are shown. The thickness of pulmonary airspaces is shown as fits of histogram distributions of voxel counts (**B**,**C**) and probabilities (**D**,**E**) per airspace diameter (µm) of the entire lung. A total of 3 ctrls (solid, blue line) and 3 βENaC-Tg (dashed, red line) lungs were analyzed (**B**,**D**). The average per group (ctrl and βENaC-Tg) is shown in (**C**,**E**). The position of the peaks, representing the most frequent airspace diameters, are labeled with black dotted lines, and the values of the peak positions are given in blue for the ctrl and red for the βENaC-Tg groups. ctrl = control.

**Figure 7 cells-12-02375-f007:**
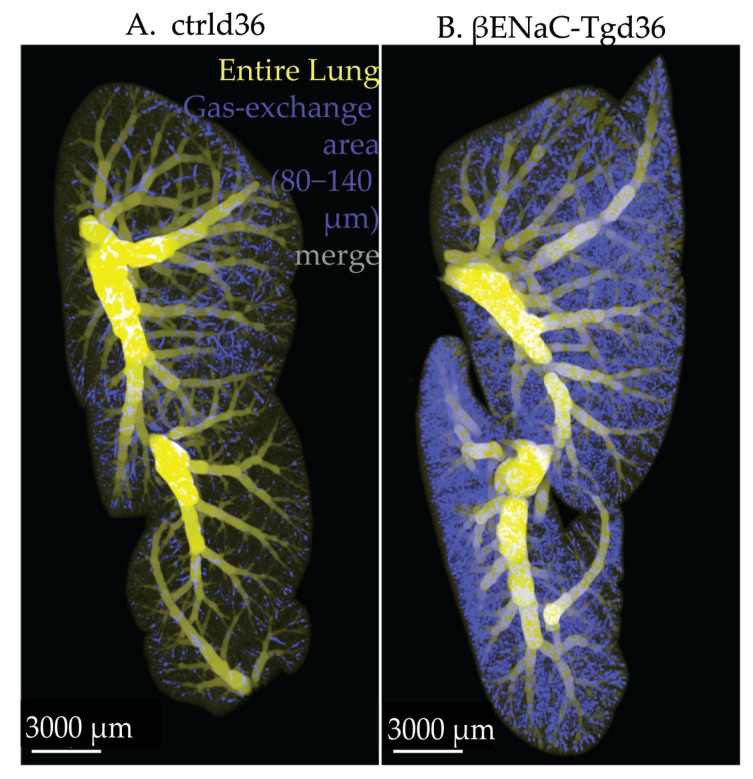
Visualization of pulmonary airspaces within the fit curve’s shoulder of control and βENaC-Tg mice from pnd36. The gas exchange area with diameter sizes ranging between 80 and 140 μm (blue) was merged into a visualization of the airspace diameter map of the entire lung (yellow) using the software Imaris. The merged image is shown for control (**A**) and for βENaC-Tg (**B**) mice. Prior to extraction of the gas exchange area (diameter range 80–140 μm), the conducting airways of the control (diameters 100 μm and above) and βENaC-Tg (diameters 140 μm and above) were subtracted from the entire lung.

**Figure 8 cells-12-02375-f008:**
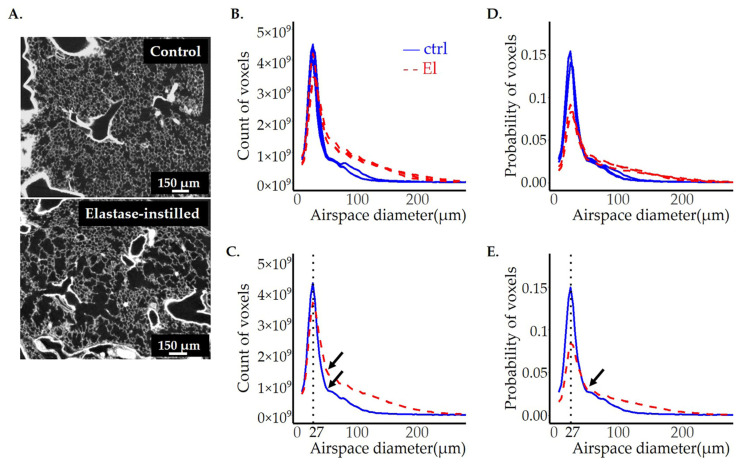
Comparison of adult mice lungs instilled with saline (ctrl) or elastase (El). One representative lung slice from the ctrl (**A**, upper panel) and El (**A**, lower panel) mice are shown. The diameter of pulmonary airspaces is shown as fits of histogram distributions of voxel counts (**B**,**C**) and probabilities (**D**,**E**) per airspace diameter (μm) of the entire lung. The total of 3 ctrls (solid, blue line) and 3 elastase (dashed, red line) lungs were analyzed and illustrated in (**B**,**D**), while the average per group (ctrl and elastase) is shown in (**C**,E). The position of the peak, representing the most frequent airspace diameter, is shown with the black dotted line, and the value of airspace diameter associated with the peak is the same in both groups and is labeled in black. The beginning of the shoulder is shown with the black arrows in (**C**,**E**).

**Figure 9 cells-12-02375-f009:**
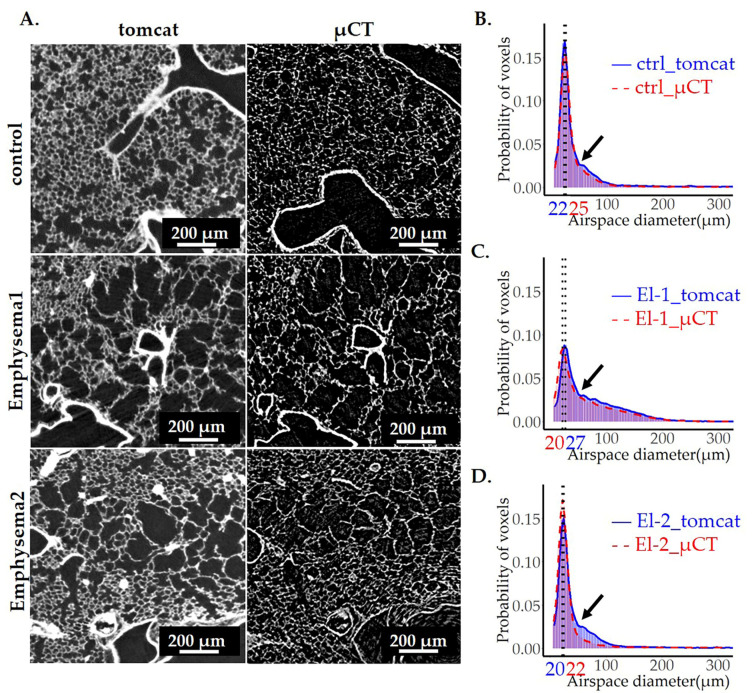
Comparison of image analyses on images obtained by µCT and SRXTM (TOMCAT). To compare the quality of results obtained by μCT to our gold standard SRXTM, three left lungs were scanned with both setups. A representative lung slice from the ctrl (upper panels) and two elastase-instilled lungs (El-1 and El-2) (middle and lower panels) imaged by SRXTM (left panels) or μCT (right panels) are shown in (**A**). The diameter of pulmonary airspaces is plotted as fits of histogram distributions of probabilities (**B**–**D**) over airspace diameter (µm). The results obtained by image analysis of SRXTM datasets are shown in blue, and the ones obtained by μCT in red. The position of the peak, representing the most frequent airspace diameter, is shown with the black dotted line. The values of airspace diameters associated with the peak are shown in blue for TOMCAT and red for the μCT. The beginning of the shoulder is labeled with the black arrows in (**B**–**D**).

**Table 1 cells-12-02375-t001:** Comparison of count and probability distributions between the conducting airways and gas exchange area for the control sample from pnd36.

					Area under the Curve ^#^		
		Peak Position (µm)	Peak Height	Peak Width at ½ Peak Height (µm)	Before the Curve’s Intersection	After the Curve’s Intersection	Volume of Airspaces (μm^3^)
Count	CA	121.6	3.6 × 10^7^	160.3	2.9 × 10^8^	1.8 × 10^9^	3.08 × 10^10^
	GEA	22.0	4.5 × 10^9^	17.3	1.4 × 10^10^	1.0 × 10^10^	3.54 × 10^11^
Probability	CA	121.6	1.4 × 10^−3^	160.3	0.02	0.06	
	GEA	22.0	0.2	17.3	0.6	0.3	

CA = conducting airways; GEA = gas exchange area (lung parenchyma); ^#^ The area under the curve is a direct measure of the airspace volumes.

**Table 2 cells-12-02375-t002:** Comparison of count and probability distributions between the ctrl and βENaC-Tg samples from pnd36.

					*Area under the Curve ^#^*		
		Peak Position (µm)	Peak Height	Peak Width at ½ Peak Height (µm)	Before the Curve’s Intersection	After the Curve’s Intersection	Volume of Airspaces (μm^3^)
Count	ctrl	22.0 (0.0)	3.8 × 10^9^ (9.5 × 10^8^)	17.1 (0.3)	1.2 × 10^10^ (3.0 × 10^9^)	1.0 × 10^10^ (2.3 × 10^9^)	3.2 × 10^11^ (7.7 × 10^10^)
	βENaC-Tg	35.0 (5.6)	4.1 × 10^9^ (2.8 × 10^8^)	30.0 (1.9)	9.2 × 10^9^ (9.4 × 10^8^)	3.6 × 10^10^ (5.8 × 10^9^)	6.6 × 10^11^ (7.3 × 10^10^)
	p	**0.02 ***	0.6	**0.0003 *****	0.2	**0.002 ****	**0.005 ****
Probability	ctrl	22.0 (0.0)	0.17 (0.003)	17.1 (0.3)	0.5 (0.01)	0.5 (0.01)	
	βENaC-Tg	35.0 (5.6)	0.09 (0.007)	30.0 (1.9)	0.2 (0.04)	0.8 (0.04)	
	p	**0.02 ***	**4.6 × 10^-5^ ******	**0.0003 *****	**0.0002n *****	**0.0002 *****	

ctrl = control; the standard deviation is shown in brackets; *p* < 0.05 *, *p* < 0.01 **, *p* < 0.001 ***, *p* < 0.0001 ****, significant *p* values are shown in bold; ^#^ The area under the curve is a direct measure of the airspace volumes.

**Table 3 cells-12-02375-t003:** Comparison of count and probability distributions between the control and elastase-instilled samples.

					Area under the Curve *^#^*		
		Peak Position (µm)	Peak Height	Peak Width at ½ Peak Height (µm)	Before the Curve’s Shoulder	Curve’s Shoulder Till End	Volume of Airspaces (μm^3^)
Count	ctrl	26.9 (0.0)	4.2 × 10^9^ (3.4 × 10^8^)	13.1 (0.2)	1.9 × 10^10^ (1.5 × 10^9^)	1.0 × 10^10^ (1.5 × 10^9^)	4.2 × 10^11^ (4.0 × 10^10^)
	El	26.9 (0.0)	3.7 × 10^9^ (4.1 × 10^8^)	18.0 (1.4)	1.8 × 10^10^ (1.7 × 10^9^)	2.5 × 10^10^ (1.3 × 10^9^)	6.3 × 10^11^ (4.1 × 10^10^)
	*p*	0.4	0.2	**0.004 ****	0.7	**0.0002 *****	**0.002 ****
Probability	ctrl	26.9 (0.0)	0.15 (0.003)	13.1 (0.2)	0.6 (0.02)	0.4 (0.02)	
	El	26.9 (0.0)	0.09 (0.005)	18.0 (1.4)	0.4 (0.02)	0.6 (0.02)	
	*p*	0.4	**6.6 × 10^−5^ ******	**0.004 ****	**0.0001 *****	**0.0001 *****	

ctrl = control; El = elastase instilled samples; The standard deviation is shown in brackets; *p* < 0.01 **, *p* < 0.001 ***, *p* < 0.0001 ****, significant *p* values are shown in bold; ^#^ The area under the curve is a direct measure of the airspace volumes.

**Table 4 cells-12-02375-t004:** Comparison of probabilities of airspace diameters obtained by image analysis of SRXTM and µCT datasets.

					Area under the Curve		
		Peak Position (µm)	Peak Height	Peak Width at ½ Peak Height (µm)	Before the Shoulder	Shoulder	After Shoulder
ctrl	SRXTM	22.0	0.17	17.7	0.7	0.2	0.1
	μCT	24.8	0.16	19.1	0.7	0.2	0.1
El-1	SRXTM	26.9	0.09	26.4	0.43	0.5	0.03
	μCT	20.3	0.08	27.8	0.46	0.5	0.03
El-2	SRXTM	22.0	0.15	19.7	0.7	**0.2**	0.1
	μCT	20.3	0.16	19.8	0.8	**0.1**	0.1

No relevant differences were observed between the SRXTM and μCT scans, except for the shoulder of the elastase-treated sample El-2 (shown with bold letters). The differences in the peak positions appear as large, but they are in the range of +/− 1 bin size of the histogram.

## Data Availability

The data are part of two larger datasets which will be published as open access datasets in the ‘PSI Public Data Repository’ (https://doi.psi.ch/detail/10.16907/ef0cfe43-0525-4e80-b77e-d6c979b6acca (accessed on 27 September 2023)) in parallel to the two still unpublished studies by the same first and last authors as this study. The tentative title of the first study is “Lung Development in a Murine Model of CF-Like Lung Disease”, and one of the second studies is “Neonatal nicotine does not have any effect on the development of an adult pulmonary emphysema”.

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
