# Peer review of "Airspace Diameter Map—A Quantitative Measurement of All Pulmonary Airspaces to Characterize Structural Lung Diseases"

_cells, 2023, doi:10.3390/cells12192375_

Round 1

Reviewer 1 Report

General comments

This paper describes a synchrotron CT imaging approach, combined with image analysis methods for determining the airspace diameter in mouse lungs. The B-ENaC mouse model of CF and the elastase model of emphysema are used to show the utility of the method. The paper also briefly describes a comparison between synchrotron acquisition and uCT.

It feels very much like the author has quickly attempted to fit a whole thesis into a paper. Overall there is too much information in this paper, and the extraneous detail should be removed and the key points should be succinctly distilled. I also suggest removing all the uCT data as it is weak by comparison to the remainder.

The major weaknesses are that there is no validation of the image processing algorithms, so it’s impossible to know whether the airway sizes reported are accurate. Given that the paper is really comparing the methodology to stereology, there should be a robust stereology analysis of these samples, but no histology images are presented as a comparison. The statistical analysis, which tests differences in curve parameters, should also be checked by a statistician. There is no sensitivity analysis, so a comparison to stereology is impossible (see specific comments).

Lots of the information in the results (e.g. all of section 3.1 to 3.3) should be in the methods as it's all information about how you did the analysis. The detail about the animal models that is included in the results should be in the introduction. The paper also needs editing for grammar and clarity.

Specific comments

Introduction

L52: Needs to mention X-ray velocimetry as a functional imaging method here. Also needs to mention PRAGMA as a way of inferring function from structure in CT scans from CF patients/animals.

L66: I wouldn't say that these are advantages. They are characteristics that can be an advantage, but the reasons why this is the case are not described.

L68: The volume of data does not have to be vast. It really depends on the application. Using 8-bit images would probably help here.

L71: I'm not sure what narrowly followed means?

Methods

L102: Did you follow the ARRIVE guidelines? If so, then state this. If not, then why not? How many animals were included in each group? Was a power calculation performed to determine the correct sample size?

L112: Why wait 1 hour before delivery? Buprenorphine is not an anaesthetic. It should really only be used for pain relief. Why deliver that, then give iso anaesthesia? Why use ket/xylazine for overdose (was it IP or IM? What dose?), that’s commonly used as an anaesthetic and has a wide therapeutic index making it less ideal for humane killing? This protocol does not make sense to me.

L120: Paper needs to discuss the limitation that this must be done in dead animals. CT, MRI, XV and flexiVent lung function testing for example, can be done in live animals. This is important, because these are the methods that you compare to at the start of the paper.

L126: This is not a particularly accurate way to measure lung volume. The flexiVent can be a better approach. But it depends what this measurement is for, and in this case the displacement method is probably adequate.

L133: Sentence should be joined to next one.

L143: I don't understand what you mean in the imaging protocol. Was an offset and a 360-degree rotation used to produce the 1.9x FOV increase in the horizontal direction? Were multiple scans performed and stitched to increase vertical field of view? If so, then this section could be made clearer because it too me a while to understand this.

L154: Should this be 50 cm?

L157: What was the resulting voxel size?

L200: Peak of what? Presumably airspace diameter? What is the value of C and D. Please provide a rationale for what they mean and why they are included. Why use the intersection point rather than peak points of each curve? What is special about the intersection?

L205: By “surface” do you mean “area”. This section should also describe count versus probability?

How have you evaluated the accuracy of the algorithm? How do you know that you're really measuring true airspace size? Was a histopathologist or radiologist involved to examine randomly selected sections?

Results

L233: I'm not sure what the information on lines 233 to 236 means? Provide further explanation.

L243: I don't understand what you mean by Ilastik projects? Why are there different numbers of projects for each group? Please clarify.

L260: Is the term particles correct here?

L281: The use of the word "over" makes it sound like you are normalising by the diameter of the airway? Why do that? Or is this just referring to the axes on the histogram?

L286: Previous chapter?

L290: How?

L292: This is one way of dividing it. You could for example also divide it into left and right?

L297: Sum of all acini? Do you mean total number or volume?

L232: What is individuum?

L306: The difference between the panels A-C is not clear. I assume that airways of diameter less than the threshold are removed?

L343: What's the difference between B and D as well as C and E? I know that you have normalised them, but why do that? Don’t they look exactly the same?

L359: What is the importance of before/after intersections?

L365: What have you done to validate these numbers?

L368: Description of CF disease should be better. In any case, this is introductory information that should not be in the results.

L376: Info should be in the methods. Is n=3 of each sufficient? Did you do a power calculation to determine this number?

L399: The rationale for using the crossing point is still not clear to me. Please describe it better (in the methods).

L410: Is it valid to do simple statistical analyses on the curve values like this? Have you consulted a statistician?

L415: Same here, this should be in the introduction

L429: It appears that the El animals are all the same severity, yet there is mention of severe disease in some and mild disease in others (L463). Why is there no variability in the graphs here?

L479: Did parameters in your algorithm require adjustment for uCT compared to SRXTM?

L481: I don't know what to make of these results. There isn't enough information here to make a proper comparison between the two techniques. I recommend removing all the uCT work, and only report on the synchrotron study.

Discussion

L505: I didn't see any rat data in here? Did you test it in rats? If not then this claim is not supported.

L508: If disease is patchy, then can your method identify that patchiness? Note that patchiness is not the same as variability in airway diameter. That doesn't appear to be mentioned here, so this claim is also not supported. Other techniques can identify local alterations in structure (PRAGMA-CT) and function (XV imaging).

L520: If disease is patchy and isolated to a small region then stereology might miss it. But so might your technique. Here in these two models you are looking at changes across the whole lung. There is no sensitivity analysis performed here to determine whether you can detect subtle disease in only a small region.

L529: It would be interesting to see how size distribution varies spatially across the lung. If you deliver elastase to only a small region of the lung do you get changes within that particular region? Are they detectable?

L535: How do you know this? Did you measure it? If not, then add a reference.

L567: I can't see this. What waves do you mean?

L568: What do you mean by fishing out?

L631: Despite reading the whole paper, it's still not clear to me what the value of knowing the distribution of total lung airspaces, rather than averages as obtained by classical methods?

The paper needs editing for grammar and clarity. 

Author Response

General comments

We thank the reviewer for the very detailed comments to our manuscript. It shows us, where we have to explain our aim, method and results more clearly and in more details. We updated the manuscript accordingly (see below) and we are sure that we have improved the manuscript. Following are the point-to-point answers to the reviewers’ comments.

This paper describes a synchrotron CT imaging approach, combined with image analysis methods for determining the airspace diameter in mouse lungs. The B-ENaC mouse model of CF and the elastase model of emphysema are used to show the utility of the method. The paper also briefly describes a comparison between synchrotron acquisition and uCT.

It feels very much like the author has quickly attempted to fit a whole thesis into a paper. Overall, there is too much information in this paper, and the extraneous detail should be removed and the key points should be succinctly distilled. I also suggest removing all the uCT data as it is weak by comparison to the remainder.

Answer: The aim of our study is to show applications of the method including pros and cons of the method. We chose two models, one very common, but somehow artificial (elastase induced pulmonary emphysema) and one less common, but very similar to a naturally accruing disease. We aimed to write the manuscript so that everyone will be able to repeat the method. Therefore, we have to report many details.

One very common critic we usually receive when we use synchrotron radiation-based X-ray tomographic microscopy is the note that only less than half of the applications for beamtime at the beamline TOMCAT will be successful and therefore our methods are only available to a small number of users. Therefore, we developed the method based on “TOMCAT”-data, but showed at the same token, that a usage of a high-resolution mCT is also possible. As conclusion, the proof of principle represents a very important part of the manuscript – and that is all we would like to show.

The major weaknesses are that there is no validation of the image processing algorithms, so it’s impossible to know whether the airway sizes reported are accurate.

Answer: Algorithm was originally developed for bone [1] and material science [2]. Our group adapted it for the analysis of lungs [3]. The current manuscript further developed the method and applied it to entire lungs (L307-309).

Given that the paper is really comparing the methodology to stereology, there should be a robust stereology analysis of these samples, but no histology images are presented as a comparison.

Answer: It was not our intention to write a manuscript comparing our method to stereology. We understand our method more as a further development of stereology than a concurrence by taking the stereological principle fully into 3-dimensions. As any method, stereology and our methods, both have shortcomings, which we believe may be discussed openly. We updated the manuscript accordingly (L-26-28, L82-85, L557-558, L568-585).

Figure 1 shows an example of our segmentation and the application of the airspace diameter map. The segmentation of every lung sample was judged by a trained stereologist by asking if the voxels were properly classified to be tissue or airspace. It is exactly the same task as point-counting to estimate the airspace or tissue volume. The only difference is that, while we count thousands of points, the stereologist counts not more than 200 points. The fraction of points for which we have to decide if it belongs to airspace or tissue is the same and represent the same challenge. To verify the airspace diameter by a stereological estimation is to our best knowledge not possible, because there is no similar parameter (L287-288).

The statistical analysis, which tests differences in curve parameters, should also be checked by a statistician. There is no sensitivity analysis, so a comparison to stereology is impossible (see specific comments).

Lots of the information in the results (e.g. all of section 3.1 to 3.3) should be in the methods as it's all information about how you did the analysis. The detail about the animal models that is included in the results should be in the introduction. The paper also needs editing for grammar and clarity.

Answer: The sections 3.1 to 3.3 describe the steps of the pipeline developed in the paper; hence, we believe that this should be part of results and not methods.

Specific comments

Introduction

L52: Needs to mention X-ray velocimetry as a functional imaging method here. Also needs to mention PRAGMA as a way of inferring function from structure in CT scans from CF patients/animals.

Answer: We thank the reviewer for this comment. We have now included both techniques in the introduction (L56-66).

L66: I wouldn't say that these are advantages. They are characteristics that can be an advantage, but the reasons why this is the case are not described.

Answer: The original sentence has been modified to:

‘This technique has the advantage of being operated with a monochromatic, high flux, partially coherent, and nearly parallel beam instead of the cone beam of a classical µCT, producing images with higher signal to noise ratio.’ (L70-73)

L68: The volume of data does not have to be vast. It really depends on the application. Using 8-bit images would probably help here.

Answer: We agree with the reviewer that the volume of data can indeed be smaller depending on the application. In our case this was not possible, as attempting the segmentation on 8-bit images resulted in much lower quality of segmentation, hence we opted for the 16-bit images. The original sentence has been modified to:

‘Furthermore, depending on the application, the produced volume of the data can be vast in which case it requires high computational power and skills as well as large storage capacity.’(L73-75)

L71: I'm not sure what narrowly followed means?

Answer: By ‘narrowly followed’, we meant that the two developments went in parallel. The original sentence has been modified to:

‘The extensive advance in the field of lung imaging techniques was followed by the development in the field of image analysis.’ (L78-79)

Methods

L102: Did you follow the ARRIVE guidelines? If so, then state this. If not, then why not? How many animals were included in each group?

Answer: Thanks for the advice. We now mention that we follow it (L124-125).

Was a power calculation performed to determine the correct sample size?

Answer: No, a power calculation was not possible, because we did not have any preliminary data before we started the experiment. Therefore, we chose the minimal necessary number of animals – as required by the Swiss authorities. We used the obtained preliminary data for the power calculation of two follow up studies – again as required by the Swiss authorities. The follow up studies are still not completed. However, most of our differences are significant, even using “n” of 3.

L112: Why wait 1 hour before delivery? Buprenorphine is not an anaesthetic. It should really only be used for pain relief. Why deliver that, then give iso anaesthesia? Why use ket/xylazine for overdose (was it IP or IM? What dose?), that’s commonly used as an anaesthetic and has a wide therapeutic index making it less ideal for humane killing? This protocol does not make sense to me.

Answer-1: We thank the reviewer for this comment and as pointed out, there is a mistake in the text. The sentence has been now modified:

Briefly, for pain prevention, 0.1 mg/kg body weight buprenorphine (Temgesic, Indivior Schweiz AG, CH) was administered subcutaneously, 1 hour before and 3 hours after vehicle/elastase instillation. Vehicle or elastase (concentration of 0.2 U/g body weight; High purity porcine pancreatic elastase, #EC134, Elastin Products, Owensville, MO, USA) were instilled intranasal under 5% isoflurane anesthesia. (L129-134)

Answer-2: The information about how mice were sacrificed was complemented and can be found in lines L134-138.

L120: Paper needs to discuss the limitation that this must be done in dead animals. CT, MRI, XV and flexiVent lung function testing for example, can be done in live animals. This is important, because these are the methods that you compare to at the start of the paper.

Answer-1: To our best knowledge, until now we are the only group which published high-resolution CT (synchrotron radiation-based X-ray tomographic microscopy) of an entire lung at a resolution to perform a segmentation of the alveolar septa vs air in a live [3,4] and a postmortem animal [5]. The study presented in this paper, which is part of two still unpublished studies, is not doable in live animals, because the imaging is so time consuming that we will never be granted the required beamtime. Furthermore, MRI does not have the required resolution. We own and use the FlexiVent. However, while the acquired data give an average over the entire lung, the detection of local differences inside the lungs is not possible.

Answer-2: The following sentences were now added to the discussion (L654-658):
“Furthermore, (iii) the experiment was done on excised lung, rather than on live animal. While different method exists for sample preparation in order to prevent degradation and motion during the image acquisition, it is reasonable to expect some alteration in the lung microstructure due to the change in surface tension. The critical point drying used here, causes sustainable shrinkage that needs to be accounted for.”

L126: This is not a particularly accurate way to measure lung volume. The flexiVent can be a better approach. But it depends on what this measurement is for, and in this case the displacement method is probably adequate.

Answer-1: Based one the weighing of interests (interest of the animal = as little suffering as possible versus interest of gain of new insides) it is not possible / not allowed in Switzerland to apply a FlexiVent measurement for the determination of the lung volume, only.

Answer-2: Scientifically, the last author grew up with stereology. He knows about the limitation of the method to determine the lung volume by water displacement. To be accurate, we separate the lobes (much better than doing it with the entire lungs) and use a dissecting needle with a tiny hook to submerse the lungs (much more accurate than forceps). The reviewer is right that we could/should repeat the description of these details in the manuscript for educational reasons (L146-148).

L133: Sentence should be joined to next one.

Answer: The sentence was joined to the next one (L155-156)

L143: I don't understand what you mean in the imaging protocol. Was an offset and a 360-degree rotation used to produce the 1.9x FOV increase in the horizontal direction? Were multiple scans performed and stitched to increase vertical field of view? If so, then this section could be made clearer because it too me a while to understand this.

Answer: Many thanks for pointing us to a text written in the jargon of the beamline. We updated the text (L164-171).

L154: Should this be 50 cm?

Answer: The propagation distance was 50 mm, there was a typo in L160, and since the same information was given 2x, we have removed the sentence in L160.

L157: What was the resulting voxel size?

Answer: the final voxel size was (1.625 μm)3. The information can be found in L163.

L200: Peak of what? Presumably airspace diameter? What is the value of C and D. Please provide a rationale for what they mean and why they are included. Why use the intersection point rather than peak points of each curve? What is special about the intersection?

Answer-1: The peak of the fit curve of airspace diameter distribution. This information was now added in the manuscript (L227).

Answer-2: Comparing two histograms, diseased versus control, we observed the following differences by eye:

  1. Position and height of the peak.
  2. Width of the peak, which we determine at half height of the peak.
  3. A shoulder of the disease lungs at the right side of the controls.
  4. An altered rising of the curves left of the intersection.

To compare the first two differences is common and straightforward. To compare difference #3 and #4 statistically, the area under the curve represents a very good measure. However, in some cases #3 and #4 were canceled out if we did not separate them. At the end, we performed #3 and #4 in order to check if the shoulder caused by the disease represents a significant alteration of not.

Figure 2 may be misleading, because the peak of the red curve is very close to the intersection. We updated the figure.

L205: By “surface” do you mean “area”. This section should also describe count versus probability?

Answer: Yes, by ‘surface’ we indeed meant ‘area’. This word was now modified in the manuscript (L233). The information about count versus probability is presented in the section that follows (2.3.3. Plotting and visualization of the distribution of enlarged airspaces).

How have you evaluated the accuracy of the algorithm? How do you know that you're really measuring true airspace size? Was a histopathologist or radiologist involved to examine randomly selected sections?

Answer-1: Because the airspace diameter map represents a true 3D-measurement, it is not possible to judge about it on 2D-sections.

Answer-2: See answer to general comments.

Results

L233: I'm not sure what the information on lines 233 to 236 means? Provide further explanation.

L243: I don't understand what you mean by Ilastik projects? Why are there different numbers of projects for each group? Please clarify.

Answer: The paragraph containing L233-236 and L243 was modified to provide further details and explanation regarding the Ilastik workflow used for tissue segmentation. The modifications can be found in L260-272 and L278-284.

L260: Is the term particles correct here?

Answer: Yes, the term ‘particles’ is correct in this context. In image analysis, the term "particle" is often used to refer to distinct objects or regions within an image that one wants to identify, measure, or analyze, and does not imply a physical particle in the conventional sense.

L281: The use of the word "over" makes it sound like you are normalizing by the diameter of the airway? Why do that? Or is this just referring to the axes on the histogram?

Answer: Indeed, it is just referring to the axes on the histogram. We updated the text. (L323)

L286: Previous chapter?

Answer: ‘Previous chapter’ has been replaced with ‘section 3.1.2.’ (L329)

L290: How?

Answer: We corrected all our measurements for shrinkage: (i) for the length measurements (e.g. airspace diameter), by dividing the value with 3√shrinkage factor (ii) for the surface measurements, by dividing the value with 2√shrinkage factor and for the (iii) volume measurements, by dividing the value with the shrinkage factor. This answer is now included in the manuscript (L333-336).

L292: This is one way of dividing it. You could for example also divide it into left and right?

Answer: We agree with the comment. The original phrase was modified with the following sentence:

The bronchial tree of the lungs may be divided into two parts (L338).

L297: Sum of all acini? Do you mean total number or volume?

Answer: We meant the total volume of all acini is equal to lung parenchyma or gas-exchange area. The sentence was modified accordingly (L343)

L232: What is individuum?

Answer: By individuum we meant the animal. The word was replaced by ‘animal’ in the manuscript. (L349)

L306: The difference between the panels A-C is not clear. I assume that airways of diameter less than the threshold are removed?

Answer: Yes, indeed, airways of diameter less than the threshold were removed, as explained in the text (L374-376). To make it more understandable, we modified the original sentence with the following:

To extract the conducting airways (CA), we first applied the above-mentioned threshold on the airway diameter map and discarded all the lower grey values (smaller airway diameters).

The panel A shows what happens when we apply a threshold that is too low (88 μm) for the extraction of the conducting airways, in which case we extract not only the conducting airways but also some of the alveoli (shown with red circles in the panel A). The panel B shows how the conducting airways look when we apply the correct threshold, and the panel C shows what happens when we apply a threshold that is too high (112 μm) for the extraction of the conducting airways, in which case we don’t extract all the conducting airways but instead we are missing some parts (as shown with white arrows). The legend for Figure 4 was slightly modified (L356-359).

L343: What's the difference between B and D as well as C and E? I know that you have normalised them, but why do that? Don’t they look exactly the same?

Answer: The Figure 5 is an illustration of the output of the separation of the gas exchange area and the conducting airways for only one sample. In this case, the only difference between B and D/C and E is the scale on the y-axes.

L359: What is the importance of before/after intersections?

Answer: See above, answer to comment to L200

L365: What have you done to validate these numbers?

Answer: See general comments

L368: Description of CF disease should be better. In any case, this is introductory information that should not be in the results.

Answer: The description for the CF disease was improved and moved to the introduction section (L102-108)

L376: Info should be in the methods. Is n=3 of each sufficient? Did you do a power calculation to determine this number?

Answer-1: The information about the numbers of animals used per condition has been moved the method section and can be found in section 2.2.1., L164 for the SRXTM experiments, and in section 2.2.2., L190-196 for the μCT experiments.

Answer-2: See above.

L399: The rationale for using the crossing point is still not clear to me. Please describe it better (in the methods).

Answer: See above, answer to comment to L200

L410: Is it valid to do simple statistical analyses on the curve values like this? Have you consulted a statistician?

Answer: See above.

L415: Same here, this should be in the introduction.

Answer: The suggested part was moved to introduction and can be found in L108-110.

L429: It appears that the El animals are all the same severity, yet there is mention of severe disease in some and mild disease in others (L463). Why is there no variability in the graphs here?

Answer: We thank the reviewer for pointing out the lack of proper explanation regarding elastase-instilled animals in this study. The elastase-instilled animals were all instilled in the same manner and the lungs shown in Figure 8, all present with the comparable damage.

For the comparison of image analysis between the two technics (SRXTM and μCT, section 3.6.) we have used one control and one elastase-instilled sample (El-1) presented in Figure 8 and we added additional sample (El-2), not previously used, that contains a milder emphysema of a still unpublished dataset to address the impact of lower exposure time (and hence higher noise) on image analysis result (this explanation was now added in the method section 2.2.2., L190-196).

L479: Did parameters in your algorithm require adjustment for uCT compared to SRXTM?

Answer: No, the algorithm was not adjusted. However, due to the lower signal to noise ratio we had to perform an additional step with ImageJ ‘analyze particles’ function in order to remove small speckles after the segmentation was done. (Described in section 3.1.2., L299-303)

L481: I don't know what to make of these results. There isn't enough information here to make a proper comparison between the two techniques. I recommend removing all the uCT work, and only report on the synchrotron study.

Answer: The comparison to mCT work is there for several reasons:

  1. To test if results obtained with the proposed image analysis pipeline on images obtained with SRXTM, can be reproduced on images with lower signal to noise ratio. We performed the comparison because we got repeatedly the critic that our methods require an imaging device that is not available to 50% of the potential users. Therefore, it is important to show as a proof of principle that a high-resolution mCT is sufficient for imaging if the right settings are chosen.
  2. To point out the importance of image quality (=adequate signal to noise ratio) in obtaining reproducible image analysis results

We completely agree that it is not possible to perform a valid statistical analysis when using only 1 or 2 samples. However, this was not the aim of this section. The aim was to apply the pipeline on both the control and the diseased samples and see if one can obtain comparable results, meaning that one can detect both the ‘regular’ size and enlarged airspaces.

We believe that we met the scope of this section and that our results point out the limitation that arise from inadequate image quality. For this reason, we would like to keep this section in the manuscript.

Discussion

L505: I didn't see any rat data in here? Did you test it in rats? If not then this claim is not supported.

Answer: The aim of this sentence was not to claim that we have reported experiments in rats (the rat data represent unpublished results), but rather to indicate the size of the lung that can be imaged with the chosen device. The statement has now been modified in the following way:

Our pipeline can be used (i) to address the distribution of airspaces throughout the entire lung and (ii) to extract and separately analyze different lung compartments. We call the results of this pipeline “airspace diameter map” (L553-556).

L508: If disease is patchy, then can your method identify that patchiness? Note that patchiness is not the same as variability in airway diameter. That doesn't appear to be mentioned here, so this claim is also not supported. Other techniques can identify local alterations in structure (PRAGMA-CT) and function (XV imaging).

L520: If disease is patchy and isolated to a small region then stereology might miss it. But so might your technique. Here in these two models you are looking at changes across the whole lung. There is no sensitivity analysis performed here to determine whether you can detect subtle disease in only a small region.

Answer-1: The reviewer raises two important points (L508 + L520). Besides one paper of our group presenting the algorithm as proof of principle [3], this is the first manuscript describing the method in detail. However, we still did not explore all of the possible applications of the method.

Answer-2: It is easy to calculate the percentage of diseased versus healthy airspace volumes based on the histograms, which we did not include in this manuscript because it is already a long manuscript.

Answer-3: We are able to visualize the enlarged airspaces. Because this is an important point, we have added the Figure 7 (L455-468). As shown in Figure 7, we are able to detect alteration of the airspace diameter by morphological observation. Based on this approach, it should be possible to detect subtle diseases in only a small region of the lung.

Answer-4: To prove that we can “detect subtle disease changes in only a small lung region”, we need a suitable disease model. In Switzerland, a “weighing of interests” (see above) is required for every animal experiment. Therefore, the chosen disease model has to account for both, weighing of interest and be suitable to answer an important scientific or clinical question. We are currently working on such a model, but the analysis is still ongoing.

L529: It would be interesting to see how size distribution varies spatially across the lung. If you deliver elastase to only a small region of the lung, do you get changes within that particular region? Are they detectable?

Answer: Indeed, this would be an interesting experiment. However, we did not perform these experiments, so it is impossible for us to answer this question. The only analysis that would be possible, with the data that we have so far, is to compare the elastase-induced damage across different lobes of the same sample and analyze whether there is variability amongst the lobes. However, this analysis lays beyond the scope of this manuscript.

L535: How do you know this? Did you measure it? If not, then add a reference.

Answer: “Morphological inspection of the lung slices of βENaC-Tg lungs and elastase-instilled lungs revealed that the damage is more homogeneously distributed in the βENaC-Tg lungs than in elastase-instilled ones” It is also visible in figures 6-9. We added this information to the manuscript (L595-597). Again, until now we did not explore all possible application of this method, but we are actively working on it.

L567: I can't see this. What waves do you mean?

Answer: We apologize to the reviewer for this typo; indeed the curves containing waves are not in Figure 5B and D as wrongly stated, but in Figure5C and E. We corrected for this typo in the text (L629).

L568: What do you mean by fishing out?

Answer: By fishing out we meant extracting from the rest of the lung. We have modified this expression with ‘extraction’ (L630).

L631: Despite reading the whole paper, it's still not clear to me what the value of knowing the distribution of total lung airspaces, rather than averages as obtained by classical methods?

Answer: We believe that knowing the distribution of total lung airspaces, rather than extrapolating from a small sample, is crucial in understanding the patterns and mechanisms of heterogeneous diseases. It is clear that science so far has not been able to adequately address the problem of lung emphysema (see L577-580) and it is therefore necessary that advances are made in this sector. Having the information about the 100% of the sample (rather less than 5%) by default provides deeper insight and higher sensitivity, because it contains the information of every individual lung airspace. The possibilities of further analysis, from there on, are numerous (see above). We can choose to analyze any particular range of airspace sizes and compare them between different conditions. In this particular manuscript, we have provided an example for separate analysis of the conducting airways and gas-exchange area (Fig. 4) and comparison of enlarged airspaces between control and βENaC-Tg mice (Fig. 7). However, of course, the possibilities do not end there. We could equally compare the volumes of any other part of the lung, for example primary/secondary/tertiary bronchi or just airspaces with a certain diameter. In combination with an imaging software, we can go one-step further and physically visualize where all the airspaces of interest are localized in the 3D space of the lung.

Furthermore, the airspace diameter maps of βENaC-Tg lungs (Fig. 6) and the ones of elastase-instilled lungs (Fig. 8), show quite a different pattern. We would have missed this difference, if we would have had only one value like the mean linear intercept per condition.

References

  1. Liu, Y.; Jin, D.; Li, C.; Janz, K.F.; Burns, T.L.; Torner, J.C.; Levy, S.M.; Saha, P.K. A robust algorithm for thickness computation at low resolution and its application to in vivo trabecular bone CT imaging. IEEE transactions on bio-medical engineering 2014, 61, 2057-2069, doi:10.1109/TBME.2014.2313564.
  2. Tran, H.; Doumalin, P.; Delisee, C.; Dupre, J.C.; Malvestio, J.; Germaneau, A. 3D mechanical analysis of low-density wood-based fiberboards by X-ray microcomputed tomography and Digital Volume Correlation. Journal of Material Science 2012, 48, 3198-3212, doi:10.1007/s10853-012-7100-0.
  3. Lovric, G.; Vogiatzis Oikonomidis, I.; Mokso, R.; Stampanoni, M.; Roth-Kleiner, M.; Schittny, J.C. Automated computer-assisted quantitative analysis of intact murine lungs at the alveolar scale. PloS one 2017, 12, e0183979, doi:10.1371/journal.pone.0183979.
  4. Lovric, G.; Mokso, R.; Arcadu, F.; Vogiatzis Oikonomidis, I.; Schittny, J.C.; Roth-Kleiner, M.; Stampanoni, M. Tomographic in vivo microscopy for the study of lung physiology at the alveolar level. Sci Rep 2017, 7, 12545, doi:10.1038/s41598-017-12886-3.
  5. Borisova, E.; Lovric, G.; Miettinen, A.; Fardin, L.; Bayat, S.; Larsson, A.; Stampanoni, M.; Schittny, J.C.; Schleputz, C.M. Micrometer-resolution X-ray tomographic full-volume reconstruction of an intact post-mortem juvenile rat lung. Histochem Cell Biol 2021, 155, 215-226, doi:10.1007/s00418-020-01868-8.

Reviewer 2 Report

This paper has done good work.

Overall, this paper presents a novel protocol for analyzing all pulmonary airspaces and provides evidence of its efficacy in characterizing structural lung diseases. The authors successfully demonstrate the distribution of airspace diameters in healthy lungs and in lungs affected by CF-like lung disease and elastase-induced emphysema. The findings suggest that the proposed airspace diameter map has the potential to contribute to the understanding and analysis of structural alterations in chronic lung diseases.

I suggest this paper to be published in present format.

Author Response

We thank the reviewer for the critical evaluation of our manuscript.